# Simulating the Effects of Different Textural Soils and N Management on Maize Yield, N Fates, and Water and N Use Efficiencies in Northeast China

**DOI:** 10.3390/plants11233338

**Published:** 2022-12-01

**Authors:** Fanchao Meng, Kelin Hu, Puyu Feng, Guozhong Feng, Qiang Gao

**Affiliations:** 1College of Resources and Environmental Sciences, Key Laboratory of Straw Comprehensive Utilization and Black Soil Conservation, Jilin Agricultural University, Changchun 130118, China; 2College of Land Science and Technology, Key Laboratory of Arable Land Conservation (North China), Ministry of Agriculture and Rural Affairs, China Agricultural University, Beijing 100193, China

**Keywords:** soil texture, fertilization, maize yield, nitrogen loss, best management practices, WHCNS model

## Abstract

Determining the best management practices (BMPs) for farmland under different soil textures can provide technical support for improving maize yield, water- and nitrogen-use efficiencies (WUE and NUE), and reducing environmental N losses. In this study, a two-year (2013–2014) maize cultivation experiment was conducted on two pieces of farmland with different textural soils (loamy clay and sandy loam) in the Phaeozems zone of Northeast China. Three N fertilizer treatments were designed for each farmland: N168, N240, and N312, with N rates of 168, 240, and 312 kg ha^−1^, respectively. The WHCNS (soil Water Heat Carbon Nitrogen Simulator) model was calibrated and validated using the observed soil water content, soil nitrate concentration, and crop biological indicators. Then, the effects of soil texture combined with different N rates on maize yield, water consumption, and N fates were simulated. The integrated index considering the agronomic, economic, and environmental impacts was used to determine the BMPs for two textural soils. Results indicated that simulated soil water content and nitrate concentration at different soil depths, leaf area index, dry matter, and grain yield all agreed well with the measured values. Both soil texture and N rates significantly affected maize yield, N fates, WUE, and NUE. The annual average grain yield, WUE, and NUE under three N rates in sandy loam soil were 8257 kg ha^−1^, 1.9 kg m^−3^, and 41.2 kg kg^−1^, respectively, which were lower than those of loam clay, 11440 kg ha^−1^, 2.7 kg m^−3^, and 46.7 kg kg^−1^. The order of annual average yield and WUE under two textural soils was N240 > N312 > N168. The average evapotranspiration of sandy loam (447.3 mm) was higher than that of loamy clay (404.9 mm). The annual average N-leaching amount of different N treatments for sandy loam ranged from 5.1 to 13.2 kg ha^−1^, which was higher than that of loamy clay soil, with a range of 1.8–5.0 kg ha^−1^. The gaseous N loss in sandy loam soil accounted for 14.7% of the fertilizer N application rate, while it was 11.1%in loamy clay soil. The order of the NUEs of two textural soils was: N168 > N240 > N312. The recommended N fertilizer rates for sandy loam and loamy clay soils determined by the integrated index were 180 and 200 kg ha^−1^, respectively.

## 1. Introduction

Maize is one of the three major grain crops, with a yield ranking first in the world [1]. Maize production in China accounts for 20% of the total maize yield and 22% of the maize planting area in the world. The maize planting area in Northeast China is about 12.48 million hectares, accounting for 32.3% of the Chinese maize planting area and 30.2% of the total maize yield [2]. In recent years, due to excessive reclamation and unreasonable cultivation, the black soil region of Northeast China has experienced serious soil degradation. The soil fertility has decreased significantly, which has seriously hindered agricultural sustainable development and also caused a negative impact on food security in China [3].

Nitrogen (N) is the main nutrient element for maize growth, and also a major factor influencing crop yield. Since the 1990s, farmers have applied excessive N fertilizer in pursuit of high yield [4]. The amount of N applied is far higher than the N demand of crops, even twice the crop N requirement in some areas [5]. A large number of experimental studies have shown that excessive N application cannot improve crop yield, but reduces N fertilizer utilization and brings about a series of environmental problems. First of all, excess N accumulates in the soil profile. Once irrigation and heavy rainfall occur, soil residual N will easily leach into the groundwater, causing the nitrate content in groundwater to exceed the drinking water standard [6]. Secondly, a large amount of fertilizer N can be lost to the atmosphere in the forms of NH_3_, N_2_O, and other gases [7]. Therefore, it is important to determine the best management practices (BMPs) for improving crop yield and N use efficiency, reducing environmental pollution and maintaining sustainable agricultural development in the black soil region of Northeast China.

Most of the black soil region in Northeast China is a typical rain-fed agricultural region. Water and nutrients are the two main factors limiting crop productivity in this region. Soil quality has a significant influence on soil water holding capacity, water availability, hydraulic conductivity, nutrient transport, and transformation processes [8], which ultimately lead to different crop yields. Generally, there are many capillary micro-pores in clay or silty clay soil, with strong water- and nutrient-holding capacities, which can lead to a surplus of sufficient water and N for crop growth [9]. However, sandy or sandy loam soil has a low water holding capacity and larger hydraulic conductivity, which easily leads to water drainage and N leaching. Soil water and solute have the characteristics of preferentially passing through macropores. With the increase in soil clay, soil residual nitrate content shows an increasing trend. The abundant meso- and micro-pores in loamy clay lead to an increased retention of nitrate. Although the availability of water and nitrate is not better than that of sandy loam, it can be kept longer in soil [10]. In addition, soil texture directly affects soil’s available water content [11], further affecting soil N transformation, crop absorption, and N loss. Studies have shown that the N mineralization rate in sandy soil is usually higher than that in silty or clay soil due to the stronger aeration of active organic matter and less physical protection. Compared with sandy loam soil, clay soil has a stronger fixation capacity for NH_4_^+^-N. The N absorption efficiency and response to N fertilizer of maize in clay soil are also more sensitive [12]. Therefore, even if the fertilizer N applied rate is the same, soil N concentration and its distribution will also vary depending on the soil texture. This may affect the relationship between N supply and maize growth under specific field conditions. Many studies have shown that the effects of soil texture on soil N transformation, crop yield, N absorption, mineral N residue, and N loss are different [13]. Therefore, soil texture is a key factor affecting crop yield and water- and N-use efficiencies (WUE and NUE).

Optimizing fertilization is helpful to improve crop yield, WUE, and NUE and reduce N loss. However, it is difficult to accurately determine the N demand of crops at different growth stages. The soil–crop system model can simulate crop growth and the dynamic of soil water and N under different climates, soil properties, and field management practices. At present, some models are used to simulate crop growth and N leaching in the black soil region of Northeast China and optimize water and N management. Liu et al. (2012) used the DSSAT-CERES-Maize model to simulate the aboveground dry matter, maize yield, and N uptake in Gongzhuling, Jilin Province, in order to better predict the maize yield [14]. Yang et al. (2013) used the DSSAT model to simulate the effects of the combined application of organic and inorganic fertilizer on maize yield, soil carbon, and N dynamics [15]. The results showed that the combined application of inorganic and organic fertilizer with a total N rate of 165 kg ha^−1^ increased the content of soil mineral N, and nitrate leaching decreased by 10 kg ha^−1^ compared with the treatment without organic fertilizer. Li et al. (2016) used the RZWQM model to simulate the effect of different textural soils on the potential yield of maize. The results showed that, under the same weather conditions, the soil textures affected the potential yield through soil moisture content, and the average yield of black soil (8 Mg ha^−1^) was higher than that of eolian sandy soil (7 Mg ha^−1^) [16]. Jiang et al. (2019) used DNDC and DSSAT models to simulate maize yield under different fertilization scenarios. The results showed that 180–210 kg ha^−1^ was more suitable for maize production in Northeast China [17]. Peng et al. (2022) used an APSIM model to evaluate and optimize the long-term impact of different soil textures on WUE and economic benefits under alfalfa–maize rotation [18]. At present, these models are mainly used to simulate the spring maize in the black soil region of Northeast China, taking into account the factors such as rain-fed conditions, straw returning, and fertilization. However, relevant studies on crop growth, N fate, and the optimization of water- and N management practices in farmlands with different soil textures are rare.

Recently, according to the characteristics of climate, soil, and field management in China, Liang et al. (2016) developed a model for the integrated water and N management of a soil and crop system (WHCNS, soil Water Heat Carbon Nitrogen Simulator) [19], which has been successfully applied to the water use, N loss, WUE, and NUE of crops in different regions such as Northwest, North and South China [20,21,22]. However, the model has not been verified and applied in the black soil area of Northeast China. Therefore, this study aims to achieve the following: (i) evaluate the applicability of the WHCNS model in simulating the fate of water and N and crop growth processes in farmland with different textural soils in the black soil region of Northeast China; (ii) simulate and analyze the effects of different textural soils and N fertilizer application on maize yield, water consumption, N fate, WUE, and NUE; and (iii) determine the reasonable amount of N fertilizer for farmlands with different textural soils by comprehensively considering agronomic effects, environmental effects, and economic benefits.

## 2. Results

### 2.1. Model Calibration and Validation

In this study, the two-year field-measured data (soil water content, nitrate N concentration, LAI, dry matter weight, and crop yield) from the N312 treatment with two textural soils were used to calibrate the model. The calibration process was carried out in the following order: soil moisture, soil N, and crop growth process. The “trial and error method” was used to adjust the model parameters until the simulated values were as consistent as possible with the measured values [23]. After calibration, all parameters were fixed, and the model was verified with measured data from other treatments. Due to space limitation, this paper only shows the comparison results of the simulated and measured values of the N312 treatment of loamy clay from 2013 to 2014, and the rest of the results are shown in the appendix (Appendix A).

The comparison results of the simulated and measured soil water content and nitrate concentration at different depths under N312 treatment in loamy clay soil are shown in Figure 1 and Figure 2, respectively. As seen in Figure 1, the peak value of soil water content was in good agreement with the rainfall events. Except for a few points, the simulated values of soil moisture in most periods agreed well with the measured values. However, the simulated soil water content of each layer in sandy loam soil fluctuates more drastically than that of loamy clay soil (Figure 1 and Appendix A), which was mainly related to the strong hydraulic conductivity and low water holding capacity of sandy loam. 

Figure 2 shows that the nitrate content of loamy clay soil changed greatly when it was above 40 cm, while the change in the 40–100 cm soil layer was relatively slight, which was related to the effects of precipitation, fertilization, and other factors on the surface soil, followed by the mineralization of organic matter in the soil and the activities of microorganisms mainly in the 0–30 cm surface soil. The nitrate concentration fluctuated with the events of fertilization, which was similar to soil water content (Figure 1 and Figure 2). In the jointing stage, urea was applied as topdressing at the rate of 208 kg ha^−1^, and the nitrate concentration in the surface soil simulated by the model showed an obvious peak at the corresponding time, indicating that the simulation results matched well with the measured values. The comparison results between the simulated and measured values of nitrate in each soil layer of the two textural soils for validation treatments are shown in the appendix (Appendix A). In general, the simulated values are in good agreement with the measured values.

Figure 3 and Figure 4 show the simulated results of LAI and total dry matter, respectively. Figure 3 shows that the model performance of LAI under N312 treatment is perfectly good. The simulated LAI of N168 treatment was lower than that of the other two treatments, which might be caused by insufficient N fertilizer supply (Appendix A). It can be seen from Figure 4 that the simulated values of total dry matter under N312 for loamy clay soil were also in good agreement with the measured values.

The *nRMSE*, E, and d of yield, TDM, and crop N uptake of calibration treatment (N312) for sandy loam were 14.26–26.56%, 0.42–0.86, and 0.76–0.98, respectively. The loamy clay were 6.61–22.56%, 0.44–0.88, and 0.79–0.96, respectively; The results of the validation treatments (N168 and N240) for sandy loam were 16.52–29.62%, 0.36–0.46, and 0.71–0.96, respectively; The values of loamy clay were 10.56–28.64%, 0.38–0.98, and 0.73–0.98, respectively. The *nRMSE* values of yield, TDM, and crop N uptake of sandy loam and loamy clay were all less than 30% (Table 1). According to the thresholds suggested by van Liew and Garbrecht [24], an acceptable simulation should have a value of *E*>0.36 and *d*>0.70. The E values of yield, TDM, and crop N uptake were all greater than 0.36, and the d values were all greater than 0.70. These results indicated that the WHCNS model performed well in simulating the yield, TDM, and crop N uptake in this black soil.

Figure 5 shows the correlation between the simulated and measured soil water content, nitrate content, yield, and LAI of the soil clay validation treatment. The determination coefficients of the four indicators reached 0.99, 0.94, 0.87, and 0.99, respectively. Their determination coefficients for sandy loam soil are 0.80, 0.80, 0.99, and 0.97, respectively (Appendix A), and their slopes of regression equations are close to 1, which indicated that the model performance was good, and the WHCNS model can be used to simulate and analyze the water consumption, N fate, and crop growth of farmlands with different textural soils in the black soil region of Northeast China.

### 2.2. Crop Yield, Water Consumption, and Water Use Efficiency under Two Textural Soils

According to the measured yields (Table 2), the yields of the three treatments of sandy loam in 2014 decreased by 30% compared with that in 2013, while those of loamy clay increased by 12%. The annual average yields of sandy loam (8257 kg ha^−1^) decreased by 28% compared with that of loamy clay (11,440 kg ha^−1^). Under the same N fertilizer rate condition, the yield of loamy clay farmland was higher than that of sandy loam. The order of the annual average yields was N240 > N312 > N168. The annual average yields of N240 (10,341 kg ha^−1^) treatment in two years were 10% and 5% higher than those of N168 (9367 kg ha^−1^) and N312 (9838 kgha^−1^), respectively (Appendix A). This shows that excessive N fertilizer application would not increase maize yield, but lead to yield reduction.

Table 2 shows the simulation results of the water balance of a 0–1 m soil profile for two textural soils and different N rates. The precipitations in 2013 and 2014 was 615 mm and 420 mm respectively. The water drainage and runoff in 2014 were much lower than that of 2013, mainly because there was no irrigation in this area, and the main water resource came from rainfall. With the increase in rainfall, the corresponding water drainage and runoff also increased. ET was the largest water consumption, accounting for 86.4% and 78.2% of the water input from sandy loam and loamy clay in farmland, respectively. The WUE of sandy loam was 1.8 kg m^−3^, and that of loamy clay was 2.8 kg m^−3^. The WUE of loamy clay was significantly higher than that of sandy loam. The order of WUE under two textural soils and different N fertilizer rates was N240 > N312 > N168.

Table 3 shows the simulated N balance in a 1 m soil profile under different N fertilizer rates. N output from farmland mainly includes crop N uptake, N leaching, and gaseous N loss. The amount of crop N uptake is the largest output of N, accounting for 54.1% and 70.6% of the N input in sandy loam and loamy clay, respectively. The amount of crop N uptake in loamy clay was significantly higher than that in sandy loam under the same N fertilizer rate (Appendix A). N leaching was mainly affected by the N fertilizer rate and rainfall, but the increasing proportion for different textural soils was different. When the N fertilizer rate increased from 168 to 312 kg ha^−1^, the N-leaching amount of sandy loam increased by 162.2%, while it increased by 173.0% for loamy clay soil. Under the same N fertilizer rate, the N-leaching amount of sandy loam increased by 6.4–16.2 kg ha^−1^ compared with loamy clay in the abundant rainfall year (2013), with an increasing ratio of 165.9−177.0%. In the normal rainfall year (2014), under the same N fertilizer rate, the N-leaching amount of sandy loam was only 0.09–0.32 kg ha^−1^ more than that of loamy clay. The annual average N-leaching amount of sandy loam was 170.3% higher than that of loamy clay, indicating that determining a reasonable N fertilizer rate for different textural soils was the key to ensuring crop yield and environmental friendly.

Gaseous N emission is another main pathway of N loss, mainly including NH_3_ volatilization and denitrification (Table 3). The NH_3_ volatilization of both textural soils increased with the increase in the N fertilizer rate. The NH_3_ volatilization amounts of the N168, N240, and N312 treatments were 21.8, 32.3 kg, and 44.3 kg ha^−1^, respectively. Compared with the N168, the amounts of N240 and N312 increased by 32.7 and 103.5%, respectively. The average NH_3_ volatilization of sandy loam was 12.2 kg ha^−1^ higher than that of loamy clay, and the total gaseous N loss in sandy loam and loamy clay accounted for 14.6 and 11.1% of N fertilizer input, respectively.

From the results of NUE, the NUE for both textural soils decreased with the increase in the N fertilizer rate: N168 > N240 > N312. The NUE was negatively correlated with total N loss and positively correlated with maize yield (Table 3). The N loss of N312 was relatively high, resulting in low NUE. In addition, there was a large difference in NUE between different years. In 2013, the NUEs of sandy loam and loamy clay were 30.2 and 40.3 kg kg^−1^, respectively, while in 2014, they were 48.8 and 50.3 kg kg^−1^, respectively.

### 2.3. Optimizing N Fertilizer Management for Two Textural Soils under Rainfed Conditions

From the simulation results, soil texture, rainfall, and fertilization management have obvious effects on maize yield, water consumption, WUE, N leaching, gaseous N loss, and NUE. In order to assess the agronomic and environmental effects, the integrated index of each treatment was calculated (Table 3). It can be seen that the N168 treatment on loamy clay and sandy loam was the best among the three N fertilizer treatments. In order to further optimize the N fertilizer rate, the normal rainfall year 2014 was selected as a case study, and different N fertilizer management scenarios were set. Under the condition of the same field management practices, a total 20 N fertilizer management scenarios for each textural soil were simulated using the WHCNS model.

The simulation results showed that with the increase in the N fertilizer rate, the maize yield for two textural soils was also increasing at first (Figure 6a); when the N fertilizer rate exceeded 180 kg ha^−1^ for sandy loam soil, the yield reached the highest, which was 7180 kg ha^−1^, while that of loamy clay reached the highest when the N fertilizer rate reached 200 kg ha^−1^, which was 11164 kg ha^−1^. The ET of loamy clay was 7.0% higher than that of sandy loam, while water drainage was 88.9% lower than that of sandy loam. WUE also showed a similar trend. WUE increased with increases in the N fertilizer rate. When the N fertilizer rate of sandy loam exceeded 180 kg ha^−1^, and that of loamy clay exceeded 200 kg ha^−1^, WUE stayed at a constant level. (Figure 6b).

With the increase in N fertilizer rate, the N leaching of sandy loam slowly increased at first, and then increased sharply (Figure 6c), while it increased slightly for loamy clay soil. Ammonia volatilization in both textural soils increased exponentially, and it was higher for sandy loam than that for loamy clay (Figure 6d). The NUE of sandy loam and loamy clay increased linearly with the N fertilizer rate at first. When the N fertilizer rate of sandy loam exceeded about 180 kg ha^−1^, and the N fertilizer rate of loamy clay exceeded about 200 kg ha^−1^, the NUE decreased rapidly with the increase in the N rate (Figure 6e).

According to the simulation results in 2014, the integrated index of 20 fertilizer N management scenarios for each textural soil was calculated (Table 4). The integrated index of sandy loam and loamy clay soils was the lowest when the fertilization rate was 40 kg ha^−1^. Under this scenario, the WUEs of sandy loam and loamy clay soils were 0.3 kg and 0.1 kg m^−3^, respectively. Their total N loss was 6.2 and 0.46 kg ha^−1^, and their crop yields only reached 17.4 and 5.0% of the maximum. Their NUEs were 16.7 and 7.9 kg kg^−1^, respectively. When the N fertilizer rate was 180 kg ha^−1^ for sandy loam soil, and it was 200 kg ha^−1^ for loamy clay soil, their WUEs were 1.8 and 2.7 kg m^−3^, respectively, and their NUEs reached 32.0 and 40.9 kg kg^−1^, respectively. The crop yields reached the maximum, and the integrated index also reached the highest, which could be recommended as BMPs of the two textural soils for maize yield in this region.

## 3. Materials and Methods

### 3.1. Experimental Site Description

Field experiments were conducted from April 2013 to September 2014 in Sankeshu (43°20′17.4″ N,124°00′29.1″ E) and Fujiajie (43°21′48.1″ N,124°05′02″0 E) experimental sites with a distance of no more than 3km, which are located in Lishu County, Jilin Province of China. The region is characterized as a semi-humid continental monsoon climate in the north temperate zone. The annual mean temperature is 12 °C and the mean annual precipitation is 531 mm; 70–80% of the annual precipitation occurs from July to September. The soil textures of Sankeshu and Fujiajie are loamy clay and sandy loam, respectively. Soil physical–chemical properties in the soil profile for each site are listed in Table 5, and soil surface chemical properties are listed in Appendix A.

### 3.2. Experiment Design

The spring maize variety Liangyu 99 was sown on 25 April 2013 and 1 May 2014 at a density of approximately 65,000 plants ha^−1^, and harvested on 5 October 2013 and 20 September 2014, respectively. On each soil texture, the field experiment was designed as a randomized block with three replicates of three fertilizer N rates: 168, 240, and 312 kg ha^−1^ by urea, which are respectively recorded as N168, N240, and N312. One-third of the fertilizer N was applied in furrows before ploughing as basal fertilizer, together with P (100 kg P_2_O_5_ ha^−1^ as Ca(H_2_PO_4_)_2_) and K (120 kg K_2_O ha^−1^ as K_2_SO_4_). The remaining N fertilizer was side-dressed in the topsoil layer at the jointing stage. Other farmland management practices were the same as local farmers.

### 3.3. Field Sampling and Laboratory Analysis

Soil profile pits were excavated to 100 cm soil depth. Soil samples for each 20 cm layer were collected before spring maize sowing. The bulk density and soil texture were measured. The soil hydraulic parameters were estimated by using the Pedo transfer function (PTF) method [25]. The 0–100 cm soil profile samples were collected at the key growth stages of spring maize (jointing stage, silking stage, filling stage, and maturity stage) for each plot, and the soil moisture content was determined by the drying method. Fresh soil samples were extracted with 1 mol L^−1^ KCl to determine the concentrations of NH_4_^+^-N and NO_3_^−^-N by using a continuous flow analyzer (AA3, Bran and Luebbe, Norderstedt, Germany).

At the same time, at the key growth stage of spring maize, both the leaf length and width were measured for all the leaves from ten representative plants in each plot to calculate leaf area index (LAI), and they were dried at 70 °C to determine the aboveground biomass. The dried plant samples were then grounded into powder to measure the N content of various organs (stems, leaves, and grains) by Kjeldahl Apparatus (KDY-9820, KETUO, China). At harvest time, the total number of cobs and total cob weight of maize were recorded in the production area. A total of 10 representative plants were selected according to the average weight of a single cob to determine the maize yield (calculated by 14% water content). Meteorological data were obtained from the local small meteorological station, mainly recording daily rainfall, maximum temperature, minimum temperature, average temperature, relative humidity, daily average wind speed, etc.

### 3.4. Model Description and Inputs

The WHCNS model includes five main modules: soil water, soil temperature, soil carbon, soil N, and crop growth. The model is driven by daily meteorological data and crop biological parameters, and computes daily processes including soil evaporation, crop transpiration, soil water movement, runoff, soil temperature, soil net mineralization, nitrification, ammonia volatilization, denitrification, and crop growth. The potential evapotranspiration is calculated using the Penman–Monteith method [26]. The water infiltration and redistribution in the soil profile were described by the Green–Ampt method [27] and Richards equation, respectively. The soil heat-transfer simulation is taken from the HYDRUS-1D model, using the convection–dispersion equation [28]. The soil C-and N-cycling concepts are taken from the DAISY model [29]. The crop model is taken from the PS123 model, which simulates crop growth processes, including crop development stage, LAI, photosynthate accumulation and allocation, maintenance respiration, and growth respiration [30]. The water stress factor is calculated by the actual transpiration divided by the potential transpiration. The N stress factor is computed based on the simulation of the crop N demand, crop N uptake, and actual soil N supply.

This model can be used to simulate and analyze the effects of various field management practices (irrigation, fertilization, mulching, tillage, straw returning, etc.) on farmland water consumption, N fate, organic matter turnover, and crop growth [31,32]. The basic input data include the latitude and altitude of the experiment site, basic physical and chemical properties of the soil, C/N ratio, field management, initial soil conditions, and daily meteorological data. For more detailed processes, see the literature [33].

### 3.5. Evaluation of Model Simulation Effect

In this study, three statistical indices were used to assess the agreement between the predicted and observed values:(i)The normalized root-mean-squared error (*nRMSE*):(1)$$nRMSE=\sqrt{\frac{1}{n}\overset{n}{\underset{i=1}{\sum }}{\left(P_{i}-O_{i}\right)}^{2}}\times \frac{100}{\bar{x}}\;\;\;\;$$(ii)Nash–Sutcliffe efficiency (*E*):(2)$$E=1-\frac{\sum_{i=1}^{n}{\left(P_{i}-O_{i}\right)}^{2}}{\sum_{i=1}^{n}{\left(O_{i}-O\right)}^{2}}\;\;$$(iii)Agreement index (*d*):(3)$$d=1-\frac{\sum_{i=1}^{n}{\left(O_{i}-P_{i}\right)}^{2}}{\sum_{i=1}^{n}{\left(\left|P_{i}-O\right|+\left|O_{i}-O\right|\right)}^{2}}\;\;\;\;\;\;\;\;\;$$
where *n* is the number of data pairs. *P_i_* and *O_i_* are the predicted and observed values, respectively. *O* is the mean of the observed data. *nRMSE* represents the percentage of the average deviation to the measured mean. *nRMSE* < 15%, *nRMSE* = 15–30%, and *nRMSE* > 30% are considered “good,” “moderate,” and “poor” agreement, respectively [31]. *E* ranges between −∞ and 1 and allows for a negative error. The d value is a descriptive measure that ranges from 0 to 1, wherein a value close to 1 indicates satisfactory model performance. Quantities of *d* ≥ 0.75 and *E* ≥ 0 are minimum threshold values for crop growth. Quantities of *d* ≥ 0.60 and *E* ≥ −1 are minimum threshold values for N-output evaluation [34].

### 3.6. Integrated Index Method

Quantitative analysis of model outputs (crop yield, water drainage, WUE, N leaching, gaseous N loss and NUE) is helpful to obtain BMPs. In this study, the integrated index method was used to obtain BMPs in the three treatments [31]. In order to maximize crop yield and economic benefits, and minimize environmental impact, the evaluation indicators include agronomy factors (*AF*), environment factors (*EF*), and value-to-cost ratio (*VCR*). Referring to the research of Hu et al. (2010), crop yield, WUE, and NUE were selected as input variables to calculate *AF* [10]. The weights of crop yield, WUE, and NUE are set to +0.6, +0.2, and +0.2, respectively, and the values of these variables are normalized to a range of 0–1. Subsequently, *AF* was calculated by summing the products of the three normalized variables and their corresponding weights. Gaseous N loss and nitrate leaching were used to calculate *EF*, and their weights were set as +0.5 and +0.5. *EF* was calculated in the same way as *AF*. The *VCR* was calculated using the following equation:(4)$$VCR=\frac{Y\times YP}{F\times FP}\;\;\;\;$$
where *F* is the amount of fertilizer (kg), *FP* is the price of the fertilizer (yuan kg^−1^), *Y* is the crop yield (kg ha^−1^), and *YP* is the price of the grain (yuan kg^−1^). The prices of the fertilizer and maize are 4.5 yuan and 2.0 yuan kg^−1^, respectively. In order to calculate the integrated index, the weights of *AF*, *EF* and *VCR* are set to + 0.7, −0.2, and +0.5 respectively [31].

### 3.7. N Fertilizer Scenarios

In order to analyze the impact of different N fertilizer management practices on crop yield and N fate under two textural soils, the selected simulation years should be able to represent the rainfall situation in most of the local years. This study took 2014 as an example, where the rainfall year type was normal, and set up different N fertilizer management practices scenarios. As it is rainfed, irrigation scenarios are not considered. The specific N fertilizer input scenarios are as follows: the range of N fertilizer rate is 20–400 kg ha^−1^, and the interval of the N fertilizer rate is 20 kg ha^−1^, so the total number of N fertilizer rates is 20 for each textural soil. Then, the calibrated WHCNS model was used to simulate crop yield, water drainage, nitrate leaching, gaseous N loss, WUE, and NUE under each scenario

## 4. Discussion

### 4.1. Effects of Soil Texture and N Management on Crop Yield, ET, and WUE

Soil texture affects the water movement and N transport, thus affecting crop yield [35]. Related researchers have shown that physical properties of soil, such as soil texture and soil water retention, can lead to differences in crop yields, such as the yield of crops in loamy soil with a relatively higher water holding capacity [36]. In this study, the average maize yield of loamy clay was 11,440 kg ha^−1^, which was significantly higher than that of 8257 kg ha^−1^ for sandy loam (Appendix A), which was consistent with other researches results [37], but lower than that of the results reported by Lu et al. [35].

Crop yields were most sensitive to precipitation variability in coarse-textured soils and less responsive to these weather parameters in medium- and fine-textured soils. Increasing SOC concentration (>2%) contributed to crop yields being less sensitive to precipitation [38]. For example, the rainfall in July 2014 was only 58.1 mm, which is a severe drought, resulting in a significant decline in the maize yield of sandy loam soil, with an annual average yield of 30% less than that of 2013. In contrast, the yield of loamy clay increased by 12% in 2014, compared with sandy loam, due to increased water retention, presumably due to increased buffering capacity against increased water loss through evapotranspiration in loamy clay. However, sandy loam has strong air permeability and lower SOC (Appendix A), and the growth of crops on sandy loam is more vulnerable to drought stress, thus affecting the yield. Huang et al. (2020) found excessive N fertilization could not achieve a higher yield in Northeast China [39]. In this study, the annual average yields of N168, N240, and N312 were 9367 kg, 10341, and 9838 kg ha^−1^, respectively (Appendix A), and their order was N240 > N312 > N168. The results showed that fertilization had an obvious effect on crop yield, but there exists a law of diminishing returns.

The hydraulic conductivity of sandy loam was high and its water holding capacity was poor, which led to higher ET and lower WUE [39]. Wang et al. (2020) found that ET and WUE under ridge furrow cultivation with medium or fine soil texture increase by 29 and 17%, respectively, compared with coarse soil [40]. Simulation results showed that water drainage was directly related to rainfall and soil texture. In 2013, the precipitation was abundant (615 mm), and ET and water drainage were the main water consumption. The average ETs of sandy loam and loamy clay were 503 and 394 mm, respectively, and their average water drainages were 157mm and 140mm, respectively. In 2014, due to the low rainfall, the difference between the water drainage and ET of the two textural soils was not obvious. The water consumption of sandy loam was higher than that of loamy clay. The main reason was that loamy clay has the stronger water holding capacity and buffer capacity to soil alternates between dry and wet, and the soil profile could be rapidly replenished after rainfall, resulting in lower ET. The loamy clay soil can improve WUE by reducing soil water drainage [12]. The difference in nutrient supply capacity caused by different soil textures also affects the WUE, and the nutrient supply capacity is positively correlated with WUE. The mineral nutrients in the loamy clay are sufficient, and its transpiration coefficient of plant is smaller, so its WUE is higher [9]. In this study, the average WUE of sandy loam is higher than that of loamy clay. The WUE of sandy loam is slightly lower than that of other research results at a nearby site, while that of loamy clay is higher (i.e., 1.9 and 2.7 kg m^−3^, respectively), mainly due to the influence of rainfall under different meteorological conditions [29].

### 4.2. Effects of Soil Texture and N Management on N Loss

Soil texture plays an important role in nitrate retention, transport, and leaching [41]. Su et al. (2014) observed that nitrate leaching was the lowest in loam soil sand highest in loamy sand [42]. Michalczyk et al. (2020) used the HERMES model to simulate the long-term effect of different textural soils on N leaching, and found that the amount of nitrate leaching was as follows: sandy loam > silty loam > clay loam, and the nitrate leaching decreased from 113.9 to 41.9 kg ha^−1^ [43]. Milroy et al. (2008) proposed that leaching varied markedly between the soil textures, and N leaching in loamy sand accounted for 20% that of acid loamy sand or sand [44]. Wang et al. (2022) pointed out that N leaching loss in the sandy-textured soils with high sand content was 1.65 times higher than that with low sand content [45]. In this study, the annual average nitrate leaching of each treatment of sandy loam is 9.3 kg ha^−1^, which is 3.4 kg ha^−1^ higher than that of loamy clay. This is due to soil texture combined with rainfall dynamics altering soil moisture dynamics, and consequently regulating soil N responses to precipitation changes. The clay loam soil more effectively buffered water stress during relatively long intervals between precipitation events, particularly after a large rainfall event. The large hydraulic conductivity of coarse texture soil and its poor ability to retain water and nutrients mean it is more likely to cause nitrate leaching [41,46]. In addition, optimizing the amount of N fertilizer for different soil textures can effectively reduce nitrate leaching. Azad et al. (2020) reported that the amount of nitrate leaching in sandy soil decreased from 122.18 to 44.98 kg ha^−1^, and loam and clay decreased from 22.1 to 13.9 kg ha^−1^ by optimized fertilization [47]. Reducing the N fertilizer rate can significantly reduce nitrate leaching. Zhang et al. (2015) showed that the 150–240 kg ha^−1^ fertilizer rate could maintain the high maize yield meanwhile reducing nitrate leaching [48]. These results in the literature are clearly supported by our findings. In our study, nitrate leaching was related to the soil texture and N application rate. The nitrate leaching of a 1 m soil profile of sandy loam and loamy clay increases with increases in the N fertilizer rate. When the N fertilizer rate of sandy loam exceeds 260 kg ha^−1^, the increase rate is fast (Figure 6). The risk of nitrate leaching is further increased under the condition of a high N fertilizer rate. The maximum amount of nitrate leaching for N312 is 25.9 kg ha^−1^. The N leaching is lower than that in North China, which may be related to the frequent irrigation in North China [49].

Gaseous N loss (NH_3_ volatilization and denitrification) is another main pathway of N loss in farmland. Ammonia volatilization in farmland is closely related to soil texture, fertilizer application rate, application method, etc. Soil texture and pH have different influences on NH_3_ volatilization. Awale et al. (2016) found that, across N amendments, cumulative NH_3_ losses from sandy loam soil ranged from 0.7 to 4.3% of applied N, and were higher than those in silty clay soil (0.1–0.4% of applied N). Cumulative N_2_O emissions did not differ between soil textures and ranged from 3.7 to 7.4% of applied N across N sources [50]. In this study, the ranges of NH_3_ volatilization for sandy loam and loamy clay were 21.8–1−56.5 and 14.5–42.2 kg ha^−1^, respectively, on average accounting for 13.4% and 8.9% of the N fertilizer rate, respectively. The main reason for that may be that loamy clay has a stronger adsorption capacity for NH_4_^+^. The results are higher than previous studies, mainly due to different types of N fertilizer [49]. We also found that NH_3_ volatilization loss in the northeast black soil region is lower than that in southern rice fields and calcareous soil in the North China Plain, which is closely related to different soil textures, soil pH, and climate conditions [51].

The effect of soil texture on denitrification is mainly through the change in soil water holding capacity, porosity, and aeration. Jamali et al. (2016) found that denitrification increased with soil texture becoming finer [52]. This is influenced by several factors such as the differences in soil physical and chemical properties, climate, and agricultural technology. Gaillard et al. (2016) found that the N_2_O emission by denitrification ranged from 74 to 98% of the total N_2_O emission, and the N_2_O emission from silty sandy loam was the highest, with an average emission raised from 80% to 158% that of loamy sand and sandy loam [53]. Pihlatieet al. (2004) found that, within a certain range, the N_2_O emission from soil was positively correlated to the soil moisture content in organic soil, clay, and loamy sand [54]. Soil moisture content is directly related to clay content, while soil aeration is negatively correlated to clay content. Soil water content plays an indirect role in regulating N_2_O emission, mainly by affecting the oxygen supply and demand in soil to control nitrification and denitrification processes. Our research confirmed this point. In this study, the denitrification emission of sandy loam is 3.65 kg ha^−1^, which is smaller than that of loam clay (6.81 kg ha^−1^). The results are consistent with the above studies. In addition, due to the difference in water content of sandy loam soil and loamy clay soil caused by rainfall (Figure 1 and Appendix A), the denitrification rate of sandy loam soil is 7.6 kg ha^−1^ lower, while in 2014, the denitrification rate decreased significantly, and the difference between the two textural soils is not obvious. Our results were also consistent with previous studies.

### 4.3. Effects of Soil Texture and N Management on N Uptake and N Use Efficiency

The soil texture and N management will result in the difference in crop N uptake. Previous studies have found that soil water content in sandy soil (0–60 cm root zone in 1–2 days after rainfall) for irrigated maize is higher than that in clay or loam soil [12,36]. However, soil water tends to infiltrate into the deeper layers rapidly, which is not conducive for maize root to absorb or utilize water and nutrients. Contrarily, a high water content in clay soil is more favorable for maize to adsorb nitrate. The accumulation of nitrate in the soil root zone is crucial to improve soil N-supplying capacity and crop N uptake [12]. In this study, the crop N uptake from 2013 to 2014 was 121.2–182.4 kg ha^−1^ in sandy loam and 174.5–249.4 kg ha^−1^ in loamy clay. The average crop N uptake of the three treatments in loamy clay was 55.04 kg N ha^−1^ higher than that in sandy loam (Table 3). In addition, except that the N240 treatment had the highest N uptake under sandy loam in 2013, the N uptake by crops increased with increases in the N fertilizer application rate, which was also consistent with the previous studies [55].

Improving NUE is a critical strategy to mitigate the impact of N loss on the environment [56]. Cao et al. (2017) optimized the N fertilizer management of maize on silt loam, reduced the N fertilizer application rate by 36%, and improved the NUE by 20–61% [57]. Zheng et al. (2016) showed that the NUE of summer maize in sandy loam soil was 33.8–47.3% [58]. Optimizing N fertilizer use is necessary to avoid the over-application of N fertilizer, high N loss, or decreased NUE. Li et al. (2007) proposed that reducing fertilizer N input can reduce N loss and improve NUE [59]. It is also possible to achieve greater NUE by adjusting the amount and effective period of fertilization for different soil textures to synchronize crop demand [60]. Sandy loam will usually lead to water drainage and nitrate leaching, thus decreasing NUE. In this study, crop N uptake and NUE were higher on loamy clay than on sandy loam (Figure 5). In addition, N loss increased with increases in N fertilizer application according to the simulation results. When the N fertilizer rate exceeds the critical value, crop yield tends to become stable. Continuing to increase N input will significantly increase N loss and reduce NUE (Figure 6).

The N fertilizer application presents inaccuracy on soil N supply capacity and the yield level of the different regions. Previous studies rarely considered the difference caused by soil textures and climate conditions in different regions An N fertilizer positioning experiment in Jilin Province showed that the N fertilizer application rate for maize should be 180–240 kg ha^−1^ under the yield level of 12,000–14,000 kg ha^−1^ maize [61]. Wu et al. (2015) recommended a 114–224 kg ha^−1^ N fertilizer application rate in the spring maize region of Northeast China [62]. However, in this study, the simulation results showed that the optimal N fertilizer rates for sandy loam and loamy clay soil were 180 and 200 kg ha^−1^, respectively. Compared with the conventional N fertilizer rate of 221 kg N ha^−1^, the N fertilizer rate decreased by 18.6 and 9.5%, respectively (Table 4), without reducing maize yield due to the reduction in the N fertilizer rate.

## 5. Conclusions

The WHCNS model was calibrated and evaluated based on field-measured data for spring maize under different N fertilizer management and two textural soils. The results indicated that the WHCNS model could be used to quantitatively analyze the maize yield, water consumption, N fates, and WUE and NUE, and recommended the BMPs for maize yield in the black soil region of Northeast China.

The simulation results showed that average maize yields, WUE, and NUE were 3984 kg ha^−1^, 0.9 kg m^−3^, and 16.1 kg kg^−1^ higher in loamy clay soil than those of sandy loam soil, respectively, while the total N loss in loamy clay soil was 2.4 kg ha^−1^ lower than that in sandy loam soil. Compared with sandy loam, maize planted in loamy clay had a higher yield, crop N uptake, WUE, and NUE, and lower N loss.

Under the same N fertilizer rate condition, the yield of loamy clay farmland was higher than that of sandy loam. The order of the annual average yields and WUE was N240 > N312 > N168, while the order of NUE of two textural soils was N168 > N240 > N312. This showed that excessive N fertilizer application would not increase maize yield, but lead to yield reduction. Therefore, the integrated index method was used to determine the BMPs for the two textural soils. The optimal N fertilizer rates in sandy loam and loamy clay soils were 180 and 200 kg ha^−1^, respectively, which not only ensured high crop yield, along with high WUE and NUE, but also reduced environmental N loss.

## Figures and Tables

**Figure 1 plants-11-03338-f001:**
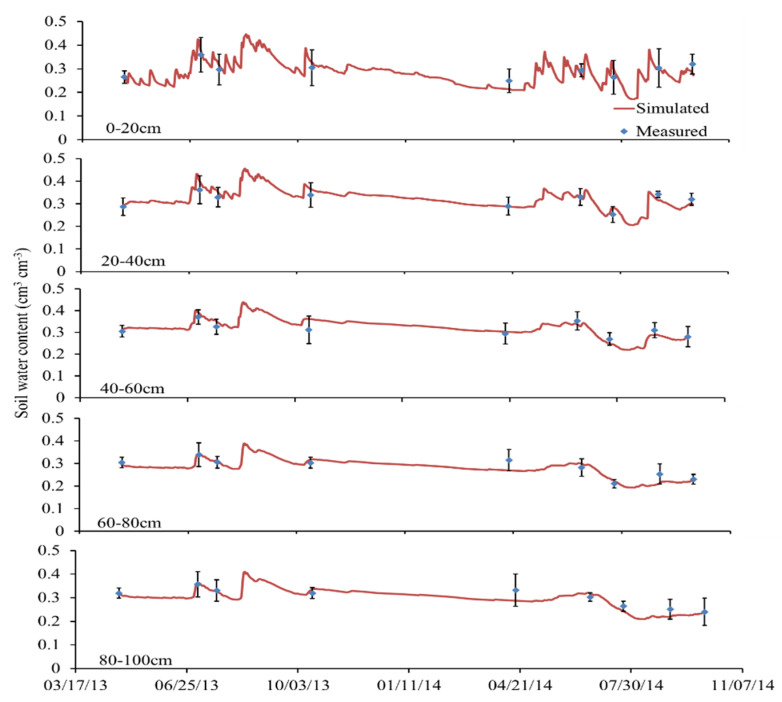
Comparison of simulated and measured soil water content at different depths under N312 treatment in loamy clay soil.

**Figure 2 plants-11-03338-f002:**
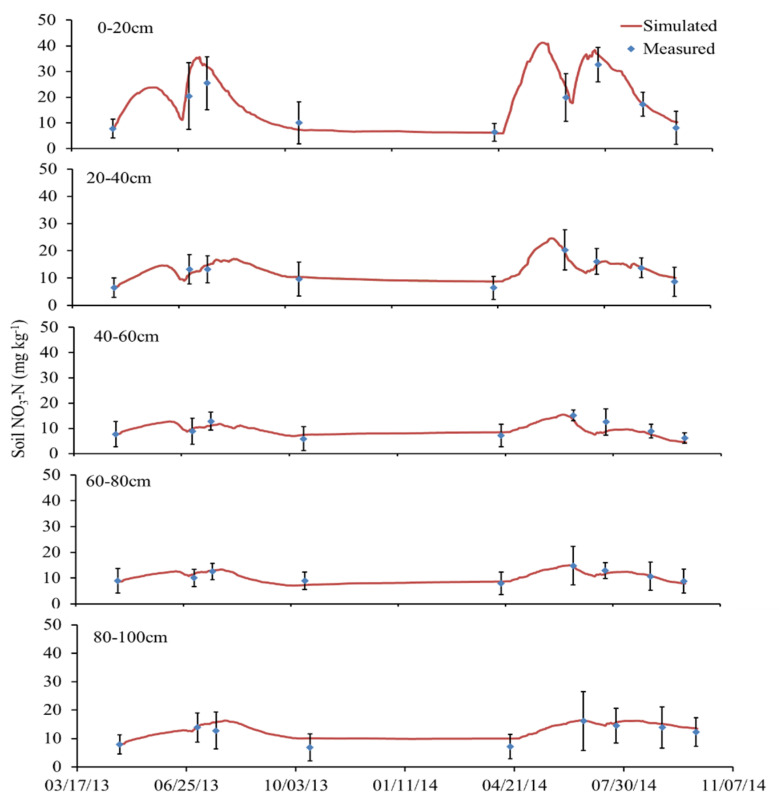
Comparison of simulated and measured soil nitrate concentration at different depths under N312 treatment in loamy clay soil.

**Figure 3 plants-11-03338-f003:**
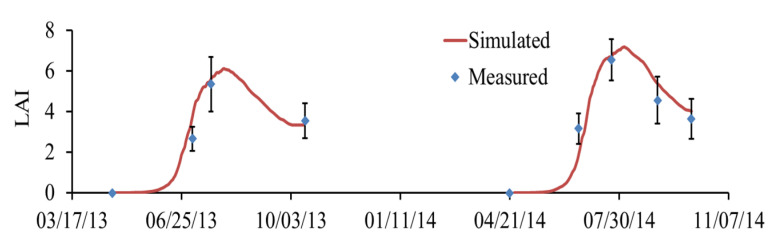
Comparison of measured and simulated leaf area index (LAI) under N312 treatment in loamy clay soil.

**Figure 4 plants-11-03338-f004:**
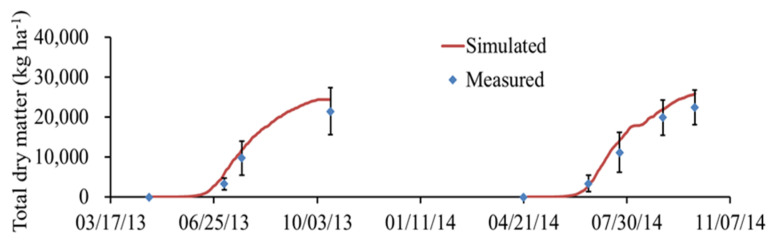
Comparison of measured and simulated total dry matter (TDM) under N312 treatment in loamy clay soil.

**Figure 5 plants-11-03338-f005:**
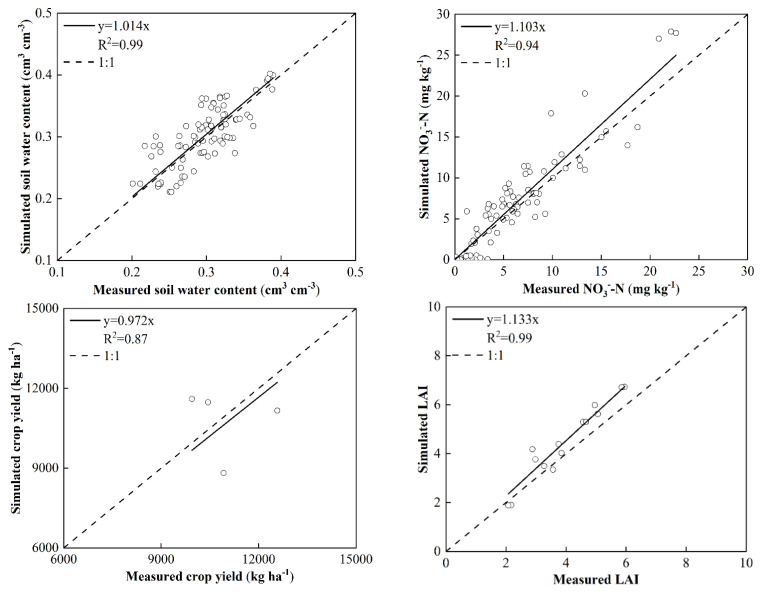
The relationship between measured and simulated soil water contents, nitrate concentrations, crop yields, and LAI under the validation treatments in loamy clay soil.

**Figure 6 plants-11-03338-f006:**
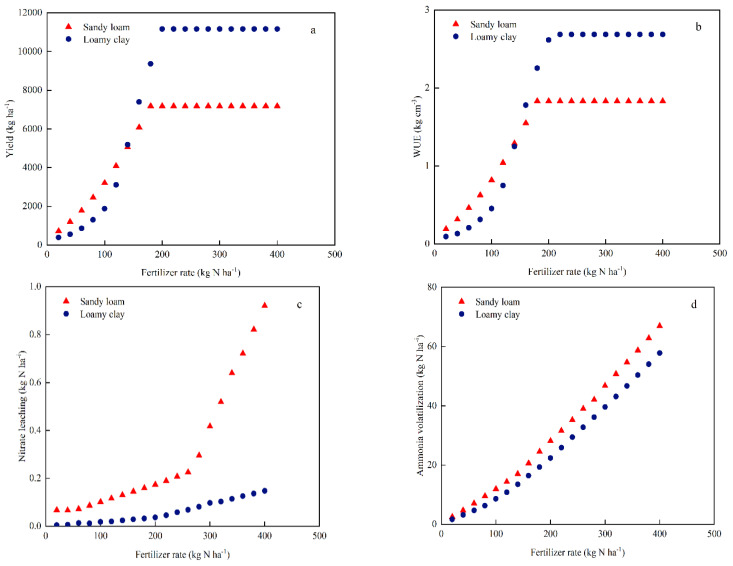
Responses of simulated maize yield (**a**), WUE (**b**), nitrate leaching (**c**), ammonia volatilization (**d**), and NUE (**e**) to different N fertilizer rates for sandy loam and loamy clay soils in 2014.

**Table 1 plants-11-03338-t001:** Statistical criteria for simulation of total dry matter, yield, and crop N uptake.

Soil Texture	Calibration	Yield			Total Dry Matter			Crop N Uptake		
	Validation	*nRMSE*	*E*	*d*	*nRMSE*	*E*	*d*	*nRMSE*	*E*	*d*
Sandy loam	C	14.26	0.42	0.76	12.16	0.86	0.98	26.56	0.56	0.86
	V	16.52	0.36	0.71	19.62	0.92	0.96	29.62	0.46	0.82
Loamy clay	C	12.18	0.44	0.79	6.61	0.88	0.96	22.56	0.68	0.92
	V	10.56	0.38	0.73	16.56	0.96	0.98	28.64	0.58	0.86

Note: C, calibration; V, validation; *nRMSE*, normalized root mean square error; *E*, Nash–Sutcliffe modeling efficiency; *d*, index of agreement.

**Table 2 plants-11-03338-t002:** Water balance and water use efficiency in 100 cm soil profiles under different N rates for two textural soils in 2013 and 2014.

Year	Soil Texture	Treatment	P	ET	D	R	Wbal	Yield	WUE
			(mm)	(mm)	(mm)	(mm)	(mm)	(kg ha^−1^)	(kg m^−3^)
2013	Sandy loam	N168	615.1	501.2	151.4	22.5	−60.0	8922	1.8
		N240	615.1	503.7	163.3	22.5	−74.3	9767	1.9
		N312	615.1	502.5	155.1	22.5	−65.0	9826	2.0
	Loamy clay	N168	615.1	400.5	142.7	64.8	7.1	9947	2.5
		N240	615.1	391.2	138.3	64.8	20.8	11,478	2.9
		N312	615.1	391.2	138.3	64.8	20.8	10,896	2.8
2014	Sandy loam	N168	420.1	392.2	6.3	4.9	16.7	6677	1.7
		N240	420.1	392.1	6.3	4.9	16.8	7541	1.9
		N312	420.1	392.2	6.3	4.9	16.7	6808	1.7
	Loamy clay	N168	420.1	415.6	0.8	4.9	−1.1	11,923	2.9
		N240	420.1	415.7	0.8	4.9	−1.2	12,576	3.0
		N312	420.1	415.7	0.8	4.9	−1.2	11,822	2.8
Mean	Sandy loam	N168	517.6	446.7	78.9	13.7	−21.7	7800	1.8
		N240	517.6	447.9	84.8	13.7	−28.8	8654	1.9
		N312	517.6	447.4	80.7	13.7	−24.2	8317	1.9
Mean	Loamy clay	N168	517.6	408.1	71.8	34.9	3.0	10,935	2.7
		N240	517.6	403.5	69.6	34.9	9.8	12,027	3.0
		N312	517.6	403.5	69.6	34.9	9.8	11,359	2.8

Note: P is precipitation; ET is evapotranspiration; D is drainage.; R is runoff; water balance, Wbal = P-ET-D-R; water use efficiency, WUE = measured yield/ET/10.3.3. N fate and N use efficiency under two textural soils.

**Table 3 plants-11-03338-t003:** N budgets, N-use efficiencies, and the integrated index under different N rates for two textural soils from 2013 to 2014.

Soil	Treat	Nfer	Nnet	Nvol	Nden	Nup	Nlea	Balance	WUE	NUE	AF	EF	VCR	II
		(kg ha^−1^)	(kg ha^−1^)	(kg ha^−1^)	(kg ha^−1^)	(kg ha^−1^)	(kg ha^−1^)	(kg ha^−1^)	(kg m^−3^)	(kg kg^−1^)				
2013SL	N168	168	83.7	28.3	4.3	140.2	9.9	68.9	1.8	48.8	0.9	0.9	23.6	1.0
	N240	240	84.2	42.8	4.5	182.4	19.1	75.4	1.9	39.3	1.0	1.2	18.1	0.9
	N312	312	83.9	56.49	4.6	180.5	25.9	128.4	1.4	36.7	0.9	1.5	14.0	0.7
2013LC	N168	168	109.6	14.5	12.4	174.5	3.6	72.6	2.5	48.5	0.9	0.9	26.3	1.0
	N240	240	108.3	21.9	11.9	201.7	7.0	105.8	2.7	47.3	1.0	1.2	21.3	0.9
	N312	312	108.3	29.2	11.9	204.7	9.7	164.7	2.8	42.6	1.0	1.5	15.5	0.8
2014SL	N168	168	16.2	21.8	1.7	121.2	0.2	39.4	1.7	46.1	0.9	0.6	17.7	1.1
	N240	240	16.2	35.3	2.7	146.4	0.2	71.6	1.9	40.9	1.0	0.9	14.0	1.0
	N312	312	16.2	49.2	4.1	171.5	0.5	103.0	2.5	30.2	0.9	1.5	9.7	0.7
2014LC	N168	168	11.5	22.4	1.5	213.3	0.1	−57.7	2.6	50.3	1.0	0.9	28.9	1.1
	N240	240	11.5	29.4	1.5	228.8	0.1	−8.4	3.0	48.4	1.0	1.2	20.7	0.9
	N312	312	11.9	42.2	1.5	249.4	0.2	30.6	2.8	40.3	0.9	1.5	15.4	0.7
Mean SL	N168	168	50.0	25.1	3.0	130.7	5.1	54.2	1.8	47.5	0.9	0.8	20.7	1.1
	N240	240	50.2	39.1	3.6	164.4	9.7	73.5	1.9	40.1	1.0	1.1	16.1	1.0
	N312	312	50.1	52.8	4.4	176.0	13.2	115.7	2.0	33.5	0.9	1.5	11.9	0.7
Mean LC	N168	168	60.6	18.5	7.0	193.9	1.9	7.5	2.6	49.4	1.0	0.9	27.6	1.1
	N240	240	59.9	25.7	6.7	215.3	3.6	48.7	2.9	47.9	1.0	1.2	21.0	0.9
	N312	312	60.1	35.7	6.7	227.1	5.0	97.7	2.8	41.5	1.0	1.5	15.5	0.8

Note: SL, sandy loam; LC, loamy clay; Nfer, N fertilizer; Nnet, net mineralization; Nvol, ammonia volatilization; Nden, N denitrification; Nup, crop N uptake; Nlea, N-leaching; balance = Nfert + Nnet −Nvol – Nden – Nup − Nlea; NUE (N use efficiency) = grain yield/(Nvol+ Nden + Nup + Nlea); WUE, water use efficiency; AF, agronomy factor; EF, environment factor; VCR, value-to-cost ratio; II, Integrated index.

**Table 4 plants-11-03338-t004:** Integrated index simulated by WHCNS model under different N rates for two soil textural soils in 2014.

Soil	Nfert (kg ha^−1^)	Nvol (kg ha^−1^)	Nden (kg ha^−1^)	Nlea (kg ha^−1^)	Yield (kg ha^−1^)	WUE (kg m^−3^)	NUE (kg kg^−1^)	VCR	II
Sandy loam	20	2.5	1.4	0.1	520	0.3	10.3	11.6	0.3
	⋮	⋮	⋮	⋮	⋮	⋮	⋮	⋮	⋮
	160	20.6	1.5	0.1	6680	1.8	31.2	18.3	1.0
	180	24.5	1.7	0.2	7180	1.8	32.0	17.3	1.1
	200	28.1	1.9	0.2	7180	1.8	30.0	16.0	1.0
	⋮	⋮	⋮	⋮	⋮	⋮	⋮	⋮	⋮
	400	66.9	2.8	1.0	7180	1.8	24.7	8.4	0.7
Loamy clay	20	1.7	1.8	0.0	391	0.1	8.6	8.7	0.2
	⋮	⋮	⋮	⋮	⋮	⋮	⋮	⋮	⋮
	180	19.3	1.5	0.0	9366	2.3	38.1	23.1	1.0
	200	22.4	1.5	0.0	11164	2.6	39.9	24.2	1.1
	220	25.9	1.5		11164	2.7	35.8	22.6	1.0
	⋮	⋮	⋮	⋮	⋮	⋮	⋮	⋮	⋮
	400	57.8	1.5	0.1	11164	2.7	27.3	12.4	0.7

Note: Nfer, N fertilizer; Nvol, ammonia volatilization; Nden, N denitrification; Nlea, N leaching; VCR value-to-cost ratio; II, Integrated index.

**Table 5 plants-11-03338-t005:** Soil physical and hydraulic properties for soil profiles in two experimental sites.

Site	Soil Layer (cm)	Particle Fraction (%)			Texture (USDA)	BD (g cm^−3^)	θ_fc_ (cm^3^ cm^−3^)	θ_wp_ (cm^3^ cm^−3^)	θ_s_ (cm^3^ cm^−3^)	k_s_ (cm d^−1^)
		Sand	Silt	Clay						
Fujiajie	0–20	70.6	15.6	11.8	Sandy loam	1.50	0.22	0.07	0.270	60.6
	20–40	82.2	9.7	8.5	Loamy sand	1.46	0.22	0.06	0.263	70.4
	40–60	87.7	5.5	6.8	Loamy sand	1.49	0.23	0.07	0.275	70.6
	60–80	83.9	11.7	6.4	Loamy sand	1.46	0.22	0.05	0.269	80.0
	80–100	80.0	12.5	7.5	Loamy sand	1.46	0.26	0.04	0.307	80.0
Sankeshu	0–20	32.8	24.1	43.1	Loamy clay	1.58	0.42	0.16	0.463	2.24
	20–40	31.5	37.2	30.3	Loamy clay	1.58	0.42	0.18	0.471	4.91
	40–60	33.0	30.9	36.1	Loamy clay	1.52	0.40	0.19	0.457	4.52
	60–80	21.7	51.1	27.2	Clay loam	1.54	0.38	0.18	0.406	22.7
	80–100	22.0	51.5	26.5	Clay loam	1.60	0.39	0.19	0.426	20.9

Note: BD, bulk density; θ_fc_, field capacity; θ_wp_, soil wilting point; θ_s_, soil saturated water content; k_s_, soil saturated hydraulic conductivity.

## Data Availability

Not applicable.

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
