# Peer review of "Simulating the Effects of Different Textural Soils and N Management on Maize Yield, N Fates, and Water and N Use Efficiencies in Northeast China"

_plants, 2022, doi:10.3390/plants11233338_

Round 1

Reviewer 1 Report

The manuscript “Simulating the effects of different textural soils and N management on maize yield, N fates and water and N use efficiencies in Northeast China” explores the response of maize to different textural soils amended with varying N levels. The study is very interesting, informative and novel. The statistical procedures used are well suitable. However, I have the following comments for its further modification.

·       The title needs to be modified as “Simulating the effects of different textural soils treated with various nitrogen levels on maize yield and N nutrition  in Northeast China”.

·       Line 14: what dose black soil region refers to?

·       Line 16: why too much high levels of N (with N rates of 168, 240 and 312kg N ha-1) were used?

·       Line 25-26: why the performance of loamy clay was better than loamy sand? With respect to annual average yield, WUE and NUE. And what do you mean by yield here (grain or biomass)?

·       Line 32-33: why the recommended N rate for sandy loam is lower than loamy clay even though the leaching and gaseous losses are more in sandy loam?

·       Line 47-52 of the introduction section opposes the heavy use of N then why you used to much high doses of N?

·       Line 120: The term “sand” needs to be replaced by “and”.  

·       In M&M section you stated that the experiment was conducted in two different filed. Now could you please provide the data regarding the physico-chemical properties of both soils? Were the N, P, K. organic matter, Lime, pH, EC, CEC etc. same in both fields? Were the climatic conditions (rain fall, solar hrs, temperature etc.) under both filed similar? You need to provide all the mentioned data for both the field.

·       The results need further modification and explanation.

·       The discussion part is poorly written. I suggest supporting it with latest references and literature.

·       The conclusion should be briefed and must contain major findings of the study.

Reviewer 2 Report

The manuscript is well written, clear and it provides new scientific evidence in the field. However, there are few aspects that demand improving. Below I list remarks that shall be considered by the author:

1.            Introduction

It is necessary to state a research hypothesis in the introduction section.

3.            Results

In general this section is well written I do have few minor remarks below

The description based on figure 2 where samples are taken from different depths in soil, shall be reevaluated. My interpretation of the results in that figure is different than the authors’ especially when it comes to changes in NO3-N content at 40 or 60cm. (rows 284-285)

In a row 289 a reference to an appropriate figure or table is missing

It is fine to include the yield from each experimental setup in kg but it would be beneficial to add the calculated difference in between experiments in kg or in \% (row 402, 404)

The authors mentions avera yield values based on individual measurement in table 3, however the average are not featured in the table. The author shall add the average values of each data to the table from each years and from different soils.

Same comment applies to tables 4,

Citation number 34 is present in the reference list but not in the manuscript.

Reviewer 3 Report

The article indicated N fertilizer effectiveness in two soil types using a simulation approach to maize cultivation. The evaluation and validation of the simulation analysis were appropriate. Therefore, I recommend publishing the article on "plants" rapidly. 

Comments

The N fates are also influenced by soil organic matter content. If possible, could you add the data and discussion for soil organic matter?

L221. Could you show the meaning of "W" and "WP"

Reviewer 4 Report

This MS showed the simulated results by N management on different soil types. This MS corresponds to research on reducing the use fo nitrogen fertilizers as part of efforts to reduce carbon emissions in agriculture in relation to the recent abnormal climate. In addition, it is an article that considers the stability of maize yield as well as the efficiency of water use.

Reviewer 5 Report

This manuscript tries to analyze the best practices for maize cultivation in terms of crop production and water and N use efficiencies on two soil types. This study is of great interest and the topic is within the scope of the journal. The manuscript is very well structured and written, and the bibliography is very up-to-date. This research applies simulation models, already developed, to the edaphic and climatic conditions of the cultivation area, in addition to making its own measurements to validate the models. This applies simulation models already developed to the edaphic and climatic conditions of the cultivation area, in addition to making its own measurements to validate the models. The adjustments of the models provided good results, from which recommendations can be extracted for cultivation practices for the benefit of productivity and sustainability.

I think that data analysis could be improved. Since 3 replicates of each N treatment were cultivated in each year and on two soil types, an ANOVA could have been performed, right? At least for the measured parameters a statistical analysis of the differences between means should be shown.

 I recommend a major revision before being accepted for publication.

 See attached file for particular and minor comments. It contains paragraphs highlighted in yellow and comments.

Reviewer 6 Report

In this manuscript, the author evaluated the model-based simulation effects of nitrogen (N) management on maize yield, and water and N use efficiencies in two different textured soils. The study objective is interesting and the findings reported in this manuscript will advance the existing knowledge on N management in soils for crops. However, I have some observations as indicated below.

1.          Line 23, ‘with them easured values’, I think there is a space problem. It should be ‘with the measured values.

2.           Line 27, there are couple of space problems- please revise those.

3.           Line 28-31, the word ‘soil’ needs to be added after ‘loamy clay’ and ‘sandy loam’.

4.           Line 118, the spacing between ‘soil’ and ‘s’ is to be omitted. The similar is true for line 120.

5.           Line 123, there are double full stops- one is to be deleted.

6.           Lines 452 and 525, spacing should be adjusted.

7.      Lines 690-693, please revise the concluding sentence to make it grammatically correct and meaningful.

Round 2

Reviewer 5 Report

The authors have taken into account the suggestions of the reviewers and the current state of the manuscript has improved significantly.